# INSIGHT-RAG: ENHANCING LLMS WITH INSIGHT-DRIVEN AUGMENTATION

## ABSTRACT

Retrieval Augmented Generation (RAG) frameworks have shown significant promise in leveraging external knowledge to enhance the performance of large language models (LLMs). However, conventional RAG methods often retrieve documents based solely on surface-level relevance, leading to many issues: they may overlook deeply buried information within individual documents, miss relevant insights spanning multiple documents, and struggle to support tasks beyond traditional question answering without significant additional customization. In this paper, we propose **Insight-RAG**, a novel framework designed to address these issues. In the initial stage of Insight-RAG, instead of using traditional retrieval methods, we employ an LLM to analyze the input query and task, extracting the underlying informational requirements. In the subsequent stage, a specialized LLM—trained on the document database—is queried to mine content that directly addresses these identified insights. Finally, by integrating the original query with the retrieved insights, similar to conventional RAG approaches, we employ a final LLM to generate a contextually enriched and accurate response. Using two scientific paper datasets, we created evaluation benchmarks targeting each of the mentioned issues and assessed Insight-RAG against traditional RAG pipeline. Our results demonstrate that the Insight-RAG pipeline successfully addresses these challenges, outperforming existing methods by up to **60 percentage points**. Supported by our comprehensive ablation studies—including the performance of each component and the quality of the identified insights—these findings suggest that integrating insight-driven retrieval within the RAG framework not only enhances performance but also broadens its applicability to tasks beyond conventional question answering. We will release our dataset and code.

## 1 INTRODUCTION

Recent advancements in large language models (LLMs) have spurred renewed interest in Retrieval Augmented Generation (RAG) frameworks (Gao et al., 2023; Fan et al., 2024). RAG has emerged as a powerful solution for mitigating inherent challenges in LLMs—such as hallucination and the lack of recent information—by integrating external document repositories with retrieval models to produce contextually enriched responses. However, conventional RAG pipelines typically rely on surface-level relevance metrics for document retrieval, which can result in several limitations: they may overlook deeply buried information within individual documents and miss relevant insights distributed across multiple documents. These shortcomings are especially detrimental in domain-specific scenarios, where the internal knowledge of LLMs is limited and accurate retrieval is critical. Moreover, traditional RAG frameworks often struggle with tasks beyond standard QA without extensive adaptation.

Traditional retrieval mechanisms often fail to capture the nuanced insights required for complex tasks (Barnett et al., 2024; Agrawal et al., 2024; Wang et al., 2024a). For example, they may overlook deeply buried details within a single document—such as subtle contractual clauses in a legal agreement or hidden trends in a business report—and may neglect relevant insights dispersed across multiple documents, like complementary perspectives from various news articles or customer reviews. Moreover, these methods may struggle with tasks beyond simple question answering, such as selecting the best job candidate from resume databases or extracting strategic recommendations from qualitative survey and review data, without significant customization.

In this paper, we propose Insight-RAG—a novel framework that refines the retrieval process by incorporating an intermediary insight extraction step (see Figure 1). In the first stage, an LLM analyzes the input query and extracts the essential informational requirements, effectively acting as an intelligent filter that isolates critical insights from the query context. This targeted extraction enables the system to focus on deeper, task-specific context. Subsequently, a specialized LLM continually pre-trained (Ke et al., 2023) with LoRA (Hu et al., 2021; Zhao et al., 2024a; Biderman et al., 2024) (CPT-LoRA) on the target domain-specific corpus leverages these identified insights to retrieve highly relevant information from the document database. Finally, the original input—now augmented with these carefully retrieved insights—is processed by a final LLM to generate a context-aware response. It is important to emphasize that the novelty of Insight-RAG lies not only in using LLMs to identify insights or in introducing a specialized Insight Retriever LLM for retrieval, but rather in the **tight coupling**

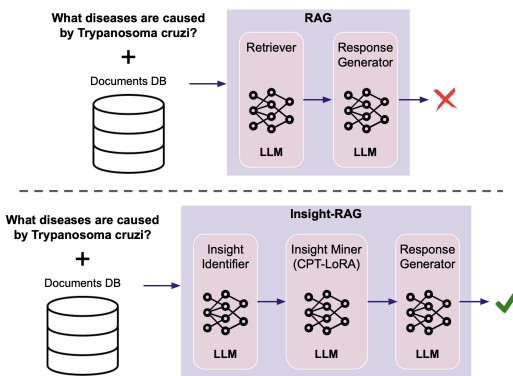

Figure 1: In conventional RAG, using a retriever model, we first retrieve relevant documents to answer a question. In contrast, in **Insight-RAG**, we first identify necessary insights to solve the task (e.g., answering a question), and then feed the identified insights to an LLM continually pre-trained over the documents to extract the necessary insights before feeding them to the final LLM to solve the task.

**between the Insight Identifier and the Insight Retriever** to extract task-relevant, contextually grounded insights that lead to substantial performance gains.

To evaluate Insight-RAG, we use two scientific paper datasets—AAN (Radev et al., 2013) and OC (Bhagavatula et al., 2018)—and create tailored datasets to address each RAG aforementioned challenge. We sample 5,000 papers from each dataset using a Breadth-First Search strategy and extract triples with GPT-4o mini (Hurst et al., 2024), followed by manual/rule-based filtering and normalization. For the deeply buried information challenge, we focus on subject-relation pairs that yield a single object, selecting only those triples where both the subject and object appear only once in each document. For the multi-document challenge, we use pairs that yield multiple objects from different documents. We then, manually filter the samples after translating each triple into a question using GPT-4o mini. Finally, for the non-QA task challenge, we use the matching labels between papers, capturing the citation recommendation task, provided by Zhou et al. (2020).

By adopting five state-of-the-art LLMs to compare Insight-RAG with the conventional RAG approach, we observe that Insight-RAG can achieve up to **60 percentage points improvement** in accuracy with much less contextual information, for both deeply buried and multi-document questions. Even against more advanced RAG variants such as **Self-RAG**, Insight-RAG still achieves substantial gains—outperforming it by up to **46 percentage points**. Moreover, we observe that for non-QA tasks such as paper matching, Insight-RAG consistently helps improve performance by up to 5.4 percentage points in accuracy, while traditional RAG shows mixed results, sometimes increasing and sometimes decreasing the performance. Then, through various ablation studies—examining the accuracy of the Insight Retriever, Insight Identifier, and the retriever in RAG baselines, as well as evaluating the quality of identified insights—we connect model behavior to the performance of different pipeline components, thereby paving the way for future applications of Insight-RAG.

## 2 INSIGHT-RAG

In this section, we detail our proposed Insight-RAG framework, which consists of three key units designed to overcome the limitations of conventional RAG approaches (see Figure 1). We define an **insight** as a task-relevant unit of information—such as a factual statement, relational triple, or data pattern—that directly addresses the informational needs of a query. Unlike surface-level textual similarity, insights capture deeper semantic connections, whether deeply embedded in a document, distributed across multiple documents, or reflected in latent patterns. Building on this definition, our

framework incorporates an intermediary insight extraction stage to capture nuanced, task-specific information that traditional methods often miss. The pipeline comprises the following units:

**Insight Identifier:**   The Insight Identifier unit processes the input to extract its essential informational requirements. Serving as an intelligent filter, it isolates critical insights from both the input and the task context, ensuring that subsequent stages concentrate on deeper, necessary content. To facilitate this process, we employ LLMs guided by a carefully designed prompt (provided in Appendix A).

**Insight Retriever:**   Inspired by prior work on insight learning (Pezeshkpour & Hruschka, 2025), the Insight Retriever serves as a specialized parametric retriever that leverages an LLM continually pre-trained on the target corpus.   Unlike common non-parametric retrievers that rely solely on embeddings, this unit internalizes domain knowledge through continual pretraining, enabling it to surface contextually grounded insights more effectively.  Specifically, we adopt Llama-3.2 3B (Grattafiori et al., 2024) as our Insight Retriever and continually pre-train it with LoRA (Zhao et al., 2024a; Biderman et al., 2024) over our scientific paper datasets. Following Pezeshkpour & Hruschka (2025), we continually pre-train the model on both the original papers and the extracted triples from them (see Section 3). This continual pre-training enables the In-

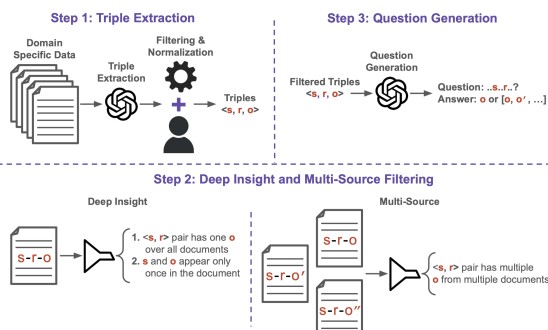

Figure 2: We create our benchmark in several steps: 1) extracting triples from domain-specific documents using GPT-4o mini and then manually normalizing/filtering them, 2) filtering the triples for each different type of issue, 3) using GPT-4o mini to translate the sampled triples to question format, asking about the object of the triple.

sight Retriever to retrieve highly relevant information that is contextually aligned with the identified requirements to solve the task.

**Response Generator:**   The final unit, response generator, integrates the original query with the retrieved insights and employs a final LLM to generate a comprehensive, context-aware response. Following the RAG approach, this augmented input allows the model to produce outputs that are both accurate and enriched by the additional insights.  The prompt for this stage is provided in Appendix A.

## 3 BENCHMARKING

To evaluate the performance of our Insight-RAG framework, we employ two scientific paper's abstract datasets—AAN and OC (provided by Zhou et al. (2020))—to create tailored evaluation benchmarks that address specific challenges encountered in conventional RAG pipelines. Figure 2 provides an overview of our process for creating the benchmarks. Below, we outline our benchmarking process for each identified issue. Data statistics are shown in Table 1, and the prompts used are provided in Appendix A.

**Deeply Buried Insight:**   In here, our focus is on the challenge of capturing deeply buried information within individual documents. We begin by sampling 5,000 papers from each dataset using a Breadth-First Search (BFS) strategy. From these papers, following previous works (Papaluca et al., 2023; Wadhwa et al., 2023), we use GPT-4o mini to extract triples (we used the same prompt provided in Pezeshkpour & Hruschka (2025)), followed by manual/rule-based filtering and normalizing the relations. Then, we select subject-relation pairs that yield a single object and ensure that both the subject and the object appear only once in the paper's abstract. This constraint guarantees that the extracted information is deeply buried and not overly prominent, thereby testing the framework's ability to capture subtle details. We then convert the curated triples into question formats using GPT-4o mini—which generates questions about the object based on the subject-relation pair—and manually filtered them for quality.

**Multi-Document Insight:** To assess the capability of Insight-RAG in synthesizing information from multiple documents, we incorporate the extracted triples from the papers. More specifically, we focus on subject-relation pairs that yield multiple objects drawn from different papers, thereby simulating scenarios where relevant insights are distributed across various documents. Once the multi-document triples are curated, we convert them into question formats using GPT-4o mini. Finally, to remove noisy and vague questions, we manually filter them to ensure quality.

**Non-QA Task:** The third benchmark addresses tasks beyond traditional question answering, specifically evaluating the framework's applicability for citation recommendation. For this benchmark, we leverage the matching labels between papers provided by Zhou et al. (2020), which capture the citation recommendation task. Our goal is to determine if the insights extracted from a document database can effectively support solving **arbitrary** tasks on inputs that share similarities with the documents, thereby extending the RAG framework's utility to a variety of real-world applications.

There are two important considerations in our benchmarking: **domain scope** and **dataset scale**. Regarding **domain**, although our evaluation is conducted on scientific paper datasets, the findings are broadly applicable to many domain-specific, in-domain settings. Scientific documents share structural and content characteristics—such as dense information, formal language, hierarchical organization, and interlinked references—with domains like legal

| | AAN | OC |
|---|---|---|
| # Docs | 5,000 | 5,000 |
| # Triples | 21,526 | 23,662 |
| # Deep-Insight Samples | 318 | 403 |
| # Multi-document Samples | 173 | 90 |
| # Matching Samples | 500 | 500 |

Table 1: Data statistics of the created benchmark.

documents, resumes, job descriptions, technical reports, financial disclosures, and medical case summaries. These similarities suggest that the challenges Insight-RAG addresses—extracting deeply buried insights, aggregating information across documents, and handling non-QA tasks—are also prevalent in many other applications. Therefore, while our benchmarks are grounded in scientific corpora, the design principles and improvements of Insight-RAG are highly generalizable. As for **scalability**, it's worth noting that only the Insight Retriever component is affected by corpus size. Prior work on LoRA-based continual pretraining (Zhao et al., 2024a; Biderman et al., 2024) has demonstrated that such approaches scale effectively to larger corpora. This supports the potential generalizability of our method beyond the size of our current datasets.

## 4 EXPERIMENTAL DETAILS

We employ several state-of-the-art LLMs as integral components of the Insight-RAG pipeline: GPT-4o, GPT-4o mini (Hurst et al., 2024), o3-mini (OpenAI, 2025), Llama3.3 70B (Grattafiori et al., 2024), and DeepSeek-R1 (Guo et al., 2025). For the Insight Retriever unit, we adopt Llama-3.2 3B as our Insight Retriever, continually pre-trained with LoRA on domain-specific scientific papers and extracted triples. We hyperparameter-tuned the Llama-3.2 3B model based on loss, with additional training and datasets details provided in Appendix B. Moreover, in the Insight-RAG pipeline, we use the same LLM for both the Insight Identifier and Response Generator. For RAG Baselines, we used LlamaIndex (Liu, 2022) and the embedding model gte-Qwen2-7B-instruct (Li et al., 2023), which is the open-sourced state-of-the-art model based on the MTEB leaderboard (Muennighoff et al., 2022) (ColBERT-based RAG results are provided in Appendix C.3). To provide a stronger baseline for comparison with Insight-RAG, we also include **Self-RAG** (Asai et al., 2023) as an extra baseline.

For fair comparison, we limit the Insight Retriever's maximum generated token length to 100 tokens for both datasets, which is less than the average document token length of 134.6 and 226.4 for AAN and OC, respectively. We observe that further increasing the maximum generated token length does not significantly change the performance. We evaluate LLM performance using accuracy, exact match accuracy (calculated by determining if the gold response exactly appears in the generated response), and F1 Score (standard QA metrics). We also employ Recall@K, which measures the proportion of correct predictions in the top-k results. We provide a detailed breakdown of the **computational complexity**, along with an in-depth discussion of the **cost implications and prompt sensitivity** of Insight-RAG, in Appendix B.

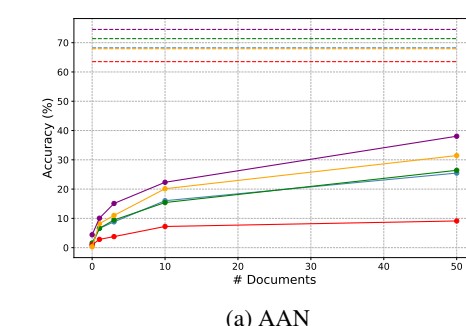

(a) AAN

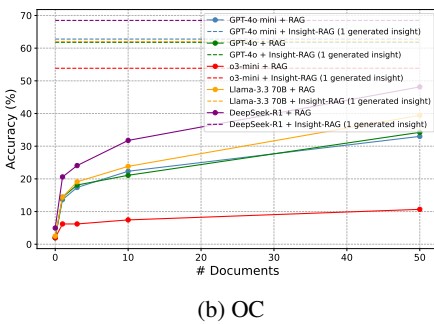

(b) OC

Figure 3: We compare RAG and Insight-RAG on the AAN and OC datasets for questions based on deeply buried information, using exact match. DeepSeek-R1 performs best, followed by Llama-3.3 70B. Insight-RAG, even with a **single** generated insight, consistently outperforms RAG by a **wide margin**, and while retrieving more documents reduces the gap, it still retains a clear advantage.

## 5 EXPERIMENTS

This section investigates the impact of Insight-RAG in addressing the aforementioned challenges: deeply buried insights, multi-document information, and non-QA tasks. We first evaluate LLMs on our benchmarks, then analyze model behavior by examining each Insight-RAG component and the quality of identified insights.

### 5.1 ANSWERING QUESTIONS USING DEEPLY BURIED INSIGHTS

Figure 3 presents the exact match accuracy of Insight-RAG versus conventional RAG using various LLMs for answering questions based on deeply buried information. First, the zero-shot performance of all LLMs—i.e., without any context or documents—is very low. This is primarily due to the domain-specific nature of the questions, which leaves the LLMs without the necessary information to solve the task. Additionally, the questions themselves may be ambiguous or even erroneous when isolated; however, providing the associated document context alleviates these issues.

As observed, Insight-RAG, even with **only one generated insight** from the Insight Retriever, achieves **significantly higher** performance compared to the conventional RAG approach. Although increasing the number of retrieved documents improves the performance of RAG, it still falls considerably short of Insight-RAG. We suspect that the shortcomings of the RAG-based solution are due to retrieval errors (see Section 5.4) and discrepancies in phrasing between the generated questions and the original text, which negatively impact performance (Modarressi et al., 2025). DeepSeek-R1 performs best, followed by Llama-3.3, both outperforming the OpenAI models. In contrast, o3 mini demonstrates the worst performance, primarily because it tends to overthink the task, which is reflected in its Insight Identifier performance (see Section 5.4).

As discussed in the Related Works section, advanced RAG variants are not directly comparable to Insight-RAG, since many of them could in principle be applied on top of it. Nevertheless, to provide a stronger baseline, we also evaluated **Self-RAG** on our benchmark (results in Appendix C.2). The results show that, while Self-RAG substantially improves standard RAG—particularly when fewer documents are retrieved—it is still **consistently and significantly (by up to 46 percentage points) outperformed by Insight-RAG**. This demonstrates that even sophisticated RAG frameworks such as Self-RAG continue to suffer from the fundamental challenges we target, underscoring superiority of Insight-RAG in addressing these issues.

We also report F1 scores of models in Appendix C.1. Surprisingly, despite the superior performance of DeepSeek in Exact Match, its performance drops significantly in F1. Upon further investigation, we observe that this is mostly due to DeepSeek's tendency to generate unnecessary content and occasional hallucinations, especially when the right document is not retrieved (we removed the thinking part of DeepSeek-generated answers to calculate the F1). Other models show similar behavior as in Exact Match, with Llama-3.3 70B emerging as the best-performing model.

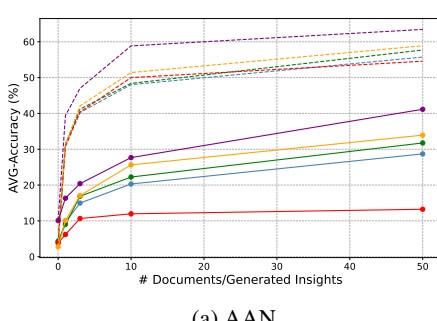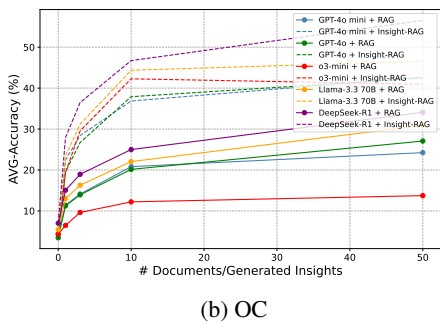

(a) AAN                                         (b) OC

Figure 4: We compare RAG and Insight-RAG on the AAN and OC datasets for multi-document questions, using averaged exact match. DeepSeek-R1 performs best, followed by Llama-3.3 70B. Insight-RAG achieves **much higher** performance with just a **few insights**, with improvements slowing as more are added.

| Model | AAN | | | OC | | |
|---|---|---|---|---|---|---|
| | Vanilla | RAG (1 doc) | Insight-RAG | Vanilla | RAG (1 doc) | Insight-RAG |
| GPT-4o mini | 80.8 | 81.6 (+0.8) | 82.8 (+2.0) | 74.4 | 70.0 (-4.4) | 78.0 (+3.6) |
| GPT-4o | 84.0 | 80.4 (-3.6) | 84.0 (0.0) | 71.6 | 73.6 (+2.0) | 74.0 (+2.4) |
| o3 mini | 85.4 | 85.6 (+0.2) | 85.6 (+0.2) | 77.0 | 74.2 (-2.8) | 82.0 (+5.0) |
| Llama 3.3 70B | 83.8 | 79.2 (-4.6) | 84.4 (+0.6) | 79.0 | 77.8 (-1.2) | 81.4 (+2.4) |
| DeepSeek-R1 | 70.4 | 74.0 (+3.6) | 73.8 (+3.4) | 66.6 | 71.4 (+4.8) | 72.0 (+5.4) |

Table 2: The performance comparison of RAG versus Insight-RAG across the AAN and OC datasets in the paper matching task, measured in terms of accuracy. As demonstrated, o3 mini performs the best while DeepSeek-R1 shows the lowest performance. Moreover, we observe that Insight-RAG consistently improves performance across all models, while RAG-based solutions show mixed impacts on model performance.

Finally, the results using **ColBERT** as the retriever (provided in Appendix C.3) closely follow the performance trends observed with the GTE model, further reinforcing our core findings on the limitations of surface-level retrieval and the advantages of insight-driven retrieval. Moreover, focusing on DeepSeek-R1 because of its superior performance, we report its RAG-based performance when, instead of retrieving documents, we **retrieve triples** from the set of all extracted triples for each dataset (see Appendix C.4). We observe that the model shows similar behavior to document-based RAG, but with much less context—since a triple is much shorter than a document—and still falls significantly short compared to Insight-RAG performance. This further highlights the shortcomings of conventional retrieval approaches and the complexity of resolving them.

## 5.2 MULTI-DOCUMENT INFORMATION AGGREGATION

We present the averaged exact match accuracy (calculated over gold answers for each sample) of Insight-RAG versus conventional RAG using various LLMs for answering questions based on information from multiple documents in Figure 4. While using the same number of retrieved documents and generated insights, Insight-RAG **consistently and significantly** outperforms the conventional RAG approach. Moreover, Insight-RAG performance increases rapidly with only a few generated insights, and then its rate of improvement slows down as more generated insights are added. While more retrieved documents improve RAG, it still lags behind Insight-RAG, though the gap narrows. Overall performance is lower in the multi-document setting than in the deeply buried case, but Insight-RAG remains clearly superior. DeepSeek-R1 leads, followed by Llama, both outperforming OpenAI models. We also report the **Self-RAG** performance, **average F1** scores, **ColBERT-based** RAG results, and **triple-based** RAG performance for DeepSeek-R1 in Appendix C. Notably, the performance trends **mirror** those observed for questions on deeply buried information. For triple-based RAG, we observe a degradation in performance—it yields results similar to document-based RAG but when using similar number of tokens in the context.

## 5.3 RAG in Non-QA Tasks

We evaluate RAG-based solutions on a non-qa task—the matching task of citation recommendation. For the RAG baseline, we retrieve only one document because the matching task is not well-defined for traditional RAG approaches, and our experiments did not show any improvement when retrieving additional documents.

Our results, presented in Table 2, show that Insight-RAG outperforms the conventional RAG baseline in nearly all settings (we report F1 results in Appendix C.1). This improvement is more pronounced on the OC dataset, likely due to the lower zero-shot performance of the LLMs on that dataset. The subjective nature of the matching task (particularly in the AAN dataset) constrains the potential for improvement, resulting in a modest performance gain. Furthermore, the RAG baseline demonstrates mixed impacts—yielding both positive and negative effects on model performance across different configurations. Notably, the o3 mini achieves the best overall performance, whereas DeepSeek-R1 performs the worst. Upon further investigation, we found that DeepSeek-R1 tends to unnecessarily overthink the task, which negatively impacts its performance. These findings underscore the effectiveness of the insight-driven approach in extending RAG to tasks beyond question answering and highlight the need for tailored retrieval strategies in non-QA contexts.

## 5.4 Components Analysis

In this section, we analyze the performance of the two key components of the Insight-RAG framework—Insight Identifier and Insight Retriever—in addition to the retriever performance of RAG baselines, and discuss how their individual contributions drive the overall success of the systems.

**Insight Identifier:** The Insight Identifier plays a crucial role by processing the input query and distilling the essential informational requirements. To measure the accuracy of the Insight Identifier for deeply buried and multi-document questions, we compare the identified insights with the gold insights (which are concatenations of the subject and relation used to generate the questions). We ask GPT-4o mini to score their similarity using a three-point scale: 0 (not similar), 0.5 (partially similar), and 1 (completely similar). We provide the prompt in Appendix A.

As shown in Figure 5, all models perform well in identifying insights for simple questions. o3 mini performs the worst, likely due to its tendency to overthink—consistent with its observed lower overall accuracy. Moreover, all models show lower performance in multi-document questions compared to deeply buried questions, which is due to the fact that when

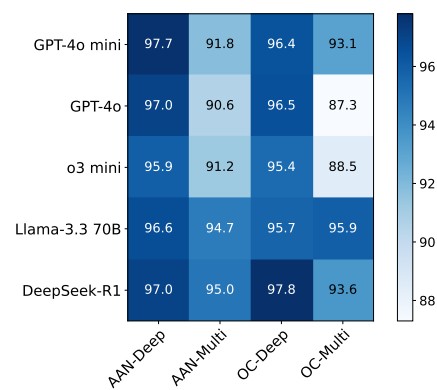

Figure 5: **Insight Identifier performance:** We ask GPT-4o mini to score the identified insights compared to the gold insights using a three-point scale: 0 (not similar), 0.5 (partially similar), and 1 (completely similar).

GPT-4o mini translates triples into question format, it tends to add more unnecessary words in multi-document questions (to capture the fact that there is more than one answer).

**Insight Retriever:** We calculate the accuracy of the Insight Retriever in predicting the object given the concatenation of subject and relation used to create questions in both deeply buried and multi-document questions. Table 3 summarizes the Insight Retriever's performance based on exact match accuracy for deeply buried questions and recall@10 for multi-document questions, respectively.

Our results show that continual pre-training of Llama3.2 3B using LoRA on both the original papers and the extracted triples leads to a reasonably well-performing Insight Retriever, with higher performance on deeply buried questions versus multi-document questions. This difference is probably due to the fact that it is easier for the model to learn information about the pair of subject and relation with one object compared to cases when there are multiple objects for a given subject-relation pair.

| Task Type | AAN | OC |
|-----------|-----|-----|
| Deep-Insight | 92.1 | 96.5 |
| Multi-Document | 72.1 | 74.8 |

Table 3: **The Insight Retriever performance:** We report exact match for deeply buried questions and Recall@10 for multiple document questions.

| Data | Deep-Insight | | Multi-Document | |
|------|--------------|-----|----------------|-------|
|      | Hits@50 | MRR | A-Hits@50 | A-MRR |
| AAN  | 39.3 | 0.13 | 46.8 | 0.16 |
| OC   | 56.1 | 0.24 | 49.5 | 0.20 |

Table 4: **The retriever performance:** We report Hits@50 and MRR for deeply buried questions, and their averages for multi-document questions.

**Retriever:** Given our knowledge of each question's source paper, we can evaluate the accuracy of the RAG baselines' retriever models in fetching relevant documents for both deeply buried and multi-document questions. Table 4 presents the retriever performance using Hits@50 and MRR metrics, along with their averaged values for multi-document questions. As shown, retriever performance is consistently low across all settings, explaining the poor performance of the RAG baselines. We attribute this low performance to two primary factors: first, embedding-based representations struggle to capture deeply buried concepts within documents; second, our question generation method produces phrasing that differs from the original text, making it harder for the retriever to find the correct document (Modarressi et al., 2025). Additionally, similar retrieval performance is observed across both settings.

## 5.5 Identified Insights in Non-QA Tasks

To better understand the identified insights and their impact on the matching task, we first extract the insights generated by the Insight Identifier module for each model and dataset. We then assign a binary label (0 or 1) to each sample, indicating whether augmenting the sample with these insights changes the model's prediction from correct to incorrect or vice versa, respectively. Next, we identify words with positive or negative impact by calculating the Z-score—a metric introduced to detect artifacts in textual datasets by measuring the correlation between the occurrence of each word and the corresponding sample label (Gardner et al., 2021). Figure 6 shows the Z-score results for the LLMs. Although in the prompt we clearly asked the models to identify insights independent of the input identifiers (i.e., Paper A and Paper B), we observe that "paper" appears as an influential token in insights identified by GPT-4o mini and o3 mini, mostly as a negative factor except for o3 mini in the OC dataset.

Overall, OpenAI models appear to benefit from relation words that indicate direct application or description (e.g., "used", "based", and "describes"), while they are hindered by more discursive or predictive terms (e.g., "presents", "discuss", "relates", and "predict"). In contrast, open LLMs perform better when relations emphasize analytical or connective processes (e.g., "analyzed", "connected", "enhance", and "involve"), with generic or usage-based terms impairing their performance (e.g., "include", "based", "used", and "applied"). This indicates that the same relation word can affect different models in opposite ways, highlighting the significant role of model architecture and training history in interpreting relational cues. Finally, we observe that for GPT-4o, most of identified insights did not result in changes to model predictions, suggesting that the Z-scores for this model may not be very trustworthy.

## 6 Related Works

RAG has emerged as a prominent strategy for enhancing LLMs by grounding their responses in external document repositories. Early works focused on improving accuracy and contextual relevance for tasks like open-domain QA and summarization by integrating retrieval mechanisms with language models (Lewis et al., 2020; Karpukhin et al., 2020; Guu et al., 2020). RAG also has been adapted for various non-QA tasks, including code generation (Wang et al., 2024b), math reasoning (Levonian et al., 2023; Yang et al., 2024), and commonsense inference (Geva et al., 2021). While these works demonstrate RAG's applicability beyond QA, they often rely on extensive customization through task-specific retrievers, dataset engineering, or other significant modifications. In contrast, Insight-RAG provides a more general-purpose solution that can be applied to diverse task–dataset pairs, requiring only the pre-training of the Insight Retriever on the target corpus (requiring no la-

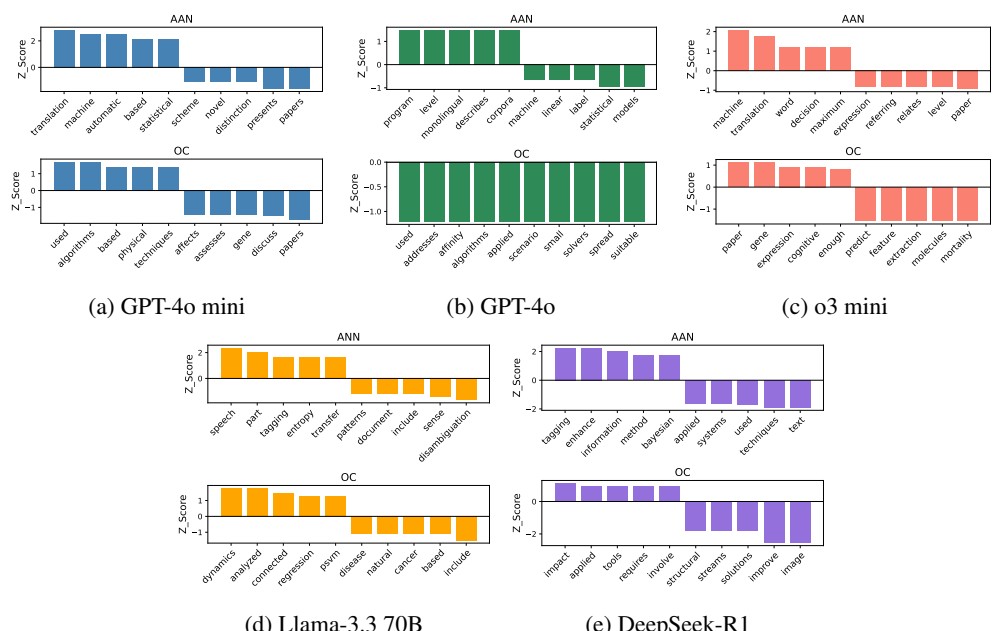

(a) GPT-4o mini  (b) GPT-4o  (c) o3 mini

(d) Llama-3.3 70B  (e) DeepSeek-R1

Figure 6: **The quality of identified insights in the matching task:** We identified the top-5 most positively and negatively influential words in the identified insights using Z-score metric.

beled data). Also, more advanced variants, such as Iter-RetGen (Shao et al., 2023) and Self-RAG (Asai et al., 2023), were proposed to handle multi-step and decomposable reasoning tasks (Zhao et al., 2024b). While not applicable to our setting of atomic, non-decomposable questions, these methods can complement Insight-RAG in more complex tasks by being applied on top of it, which we leave as future work. Moreover, recent work has explored fine-tuning LLMs to enhance specific aspects of RAG—Zhang et al. (2024) focus on domain relevance, Song et al. (2024) on hallucination suppression, and Wu et al. (2025) on dynamic retrieval routing.

In parallel to these developments, research on insight extraction has shown the value of identifying critical, often overlooked details within documents. Transformer-based methods like OpenIE6 (Kolluru et al., 2020) have advanced Open Information Extraction by using pre-training to capture nuanced relational data from unstructured text. LLMs have also emerged as powerful tools for keyphrase extraction (Muhammad et al., 2024), and in recent years, they have been increasingly adopted to mine insights from documents across various domains (Ma et al., 2023; Zhang et al., 2023; Schilling-Wilhelmi et al., 2024).

## 7 CONCLUSION AND FUTURE WORK

We introduced Insight-RAG, a novel framework that enhances traditional RAG by incorporating an intermediary insight extraction process. Our approach specifically addresses key challenges in conventional RAG pipelines—capturing deeply buried information, aggregating multi-document insights, and solving tasks beyond QA. Evaluation on our developed benchmarks from AAN and OC datasets shows that insight-driven retrieval consistently boosts performance. Moreover, through detailed ablation studies, we further identified both the reasoning behind Insight-RAG's superior performance and the shortcomings of standard RAG.

Looking ahead, Insight-RAG opens several promising research directions: (1) extending beyond citation recommendation to domains such as legal analysis, medical research, business intelligence, and creative content generation; (2) developing hierarchical insight extraction methods that categorize insights by importance, abstraction level, and relevance, to support more nuanced retrieval; (3) enabling multimodal insight extraction from text, images, audio, and video, to create a more comprehensive understanding of complex information ecosystems; (4) incorporating expert feedback loops to guide extraction in specialized fields; and (5) exploring the transferability of insights across domains to reduce the need for domain-specific training while maintaining high performance.

## REPRODUCIBILITY STATEMENT

We have taken several steps to ensure the reproducibility of our work. The design principles, methodology, and evaluation setup for Insight-RAG are detailed in Sections 2, 3, 4, and Appendix B. All models used in the experiments are well-documented, and the corresponding prompt templates are provided in Appendix A. Upon acceptance, we will release the benchmark, codebase, and full documentation covering both the experimental setup and benchmark construction to facilitate replication.

## ETHICS STATEMENT

We made use of AI tools, such as ChatGPT, to support coding tasks and assist in refining the writing of this paper. All outputs produced with AI assistance were carefully reviewed, revised, and edited by us to ensure accuracy, consistency, and alignment with our research objectives. The final content of the paper reflects our own intellectual contributions, with AI tools serving only as supportive aids in the coding and writing process.

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

## A  PROMPTS

The prompts used for the Insight Identifier, question answering with and without augmentation, matching with and without augmentation, and evaluating the identified insights are provided in prompts A.1, A.2, A.3, A.4, A.5, and A.6, respectively.

```
Insight Identifier

You are given a question or task along with its required input.  Your
goal is to extract the necessary insight that will allow another
autoregressive LLM|pretrained on a dataset of scientific papers|to
complete the answer.  The insight must be expressed as a sentence
fragment (i.e., a sentence that is meant to be completed).

Instructions:

Extract the Insight:
Identify the key information needed from the dataset to solve the task
or answer the question.
Format the insight as a sentence fragment that can be completed by the
LLM trained on the dataset.
For example, if the task is to find the birthplace of Person X, your
insight should be:  "Person X was born in".

Determine Answer Multiplicity:
Determine whether the answer should be singular or plural based solely
on the plurality of the nouns in the question.  Do not use common
sense or external context|rely exclusively on grammatical cues in the
question.
For instance, if the question uses plural nouns (e.g., "What are the
cities in California?"), set Multi-answer to True.  Conversely, if the
question uses singular nouns (e.g., "What does pizza contain?"), set it
to False.

Relevance Check:
Only include insights that are directly answerable from the dataset.
If an insight does not relate to the available dataset, ignore it.

Output Format:
Return the result as a list of dictionaries.
Each dictionary must have two keys:
"Insight":  The sentence fragment containing the key insight.
"Multi-answer":  A Boolean (True or False) indicating whether multiple
answers are required.
Example Output for follwing questions, Where was Person X born in?
what does pizza contain?  What are the Cities in California?:

[
{"Insight":  "Person X was born in", "Multi-answer":  false},
{"Insight":  "Pizza contains", "Multi-answer":  false},
{"Insight":  "The cities in California are", "Multi-answer":  true}
]

Please provide your final answer in this JSON-like list-of-dictionaries
format with no additional commentary.
Also, make sure to NOT add any extra word to the insights other than
the word present in the input.
Remove all unnecessary words and provide the insight in its simplest
form.  For example, if the query asks "what are the components that X
uses?", the insight should be "X uses".  Similarly, if the query asks
"what are all the components/techniques/features/applications included
in Z?", the insight should be "Z include".
If a non-question task is given, possible insights might involve asking
```

```
about how two concepts are connected or a definition of a concept.
Only identify the insight you believe will help solve the task, and
provide it as a short sentence fragment to be completed.  Do not add
any unnecessary content or summaries of the input.
Additionally, for non-question tasks, the insight should NOT refer to
the specific input or include any input-specific identifiers.  Instead,
it should be a STAND-ALONE statement focusing on the underlying
concepts, entities, and their relationships from the inputs.  If you
cannot find any such insights, return a list of EMPTY dictionary.

Task:
{}
```

```
QA

Answer the question.  Do not include any extra explanation.
Question:  {}
```

```
Augmented QA

Answer the question using the context.  Do not include any extra
explanation.
Question:  {}
Context:  {}
```

```
Matching

You are provided with two research papers, Paper-A and Paper-B. Your
task is to determine if the papers are relevant enough to be cited by
the other. Your response must be provided in a JSON format with two
keys:
"explanation":  A detailed explanation of your reasoning and analysis.
"answer":  The final determination ("Yes" or "No").

Paper-A:
{}

Paper-B:
{}
```

```
Augmented Matching

You are provided with two research papers, Paper-A and Paper-B, and
some useful insights.  Your task is to determine if the papers are
relevant enough to be cited by the other.  You may use the insights to
better predict whether the papers are relevant or not.  The insights
should only serve as supportive evidence; do not rely on them blindly.
Your response must be provided in a JSON format with two keys:
"explanation":  A detailed explanation of your reasoning and analysis.
"answer":  The final determination ("Yes" or "No").

Paper-A:
{}

Paper-B:
{}

Useful insights:
{}
```

```
Identified Insights Evaluation

You are given two incomplete sentences:  a target sentence and a
generated sentence.  Your task is to evaluate how similar these two
incomplete sentences are in terms of meaning and content.  Please
follow these instructions:

Similarity Criteria:

0:  The sentences are not similar at all.
0.5:  The sentences share some elements or meaning, but are only
partially similar.
1:  The sentences are very similar or essentially equivalent in
meaning.

Output Requirement:

Provide only the similarity score (0, 0.5, or 1) as your output.
Do not include any additional text or explanation.  The output format
should be as follownig:

Score:  <0, 0.5, or 1)>

Target Sentence:  {}
Generated Sentence:  {}
```

## B  EXPERIMENTAL DETAILS

**Benchmarking:**   We use the processed abstracts from the AAN dataset (Radev et al., 2013) and the OC dataset (Bhagavatula et al., 2018), as provided by Zhou et al. (2020). This curated set includes approximately 13,000 paper abstracts from AAN and 567,000 abstracts from OC, offering a rich and diverse corpus of academic content. Specifically, the AAN dataset comprises computational linguistics papers published in the ACL Anthology from 2001 to 2014, along with their associated metadata, while the OC dataset encompasses approximately 7.1 million papers covering topics in computer science and neuroscience.

**Modeling:**   For Insight Retriever, we perform continual pre-training on Llama-3.2 3B with LoRA and optimize hyperparameters through grid search based on training loss. Specifically, following Pezeshkpour & Hruschka (2025), we tuned learning rate $\alpha = [3 \times 10^{-3}, 10^{-3}, 3 \times 10^{-4}, 10^{-4}, 3 \times 10^{-5}, 10^{-5}]$; the LoRA rank $r = [4, 8, 16]$; the LoRA-alpha $\in \{8, 16, 32\}$; and the LoRA-dropout $\in \{0.05, 0.1\}$. We trained the Llama model for 30 epochs.

**Cost and Complexity Considerations:**   Continual pre-training of the Insight Retriever using LoRA on 8 NVIDIA A100 SXM GPUs for 30 epochs per dataset takes approximately 7 hours. Since Insight-RAG is intended for domain-specific settings, we argue that the initial pre-training phase constitutes the most computationally intensive step of the pipeline. While periodic updates to the Insight Retriever may be needed to accommodate newly added documents, in many real-world scenarios these updates can be performed infrequently, especially in relatively stable domains. In more dynamic environments, an online learning-based solution (Hoi et al., 2021; Liang et al., 2024) can be adopted to update the model incrementally without necessitating a full retraining cycle, thereby mitigating the associated cost.

Regarding prompting costs, while the multi-stage design of Insight-RAG introduces additional computational complexity and potential latency, which may limit its applicability in real-time or resource-constrained settings, it also enables significantly higher performance using much shorter context lengths, ultimately reducing overall API costs compared to conventional RAG. Furthermore, although the framework relies on carefully crafted prompts for the Insight Identifier, our prompt development experiments showed that downstream performance was only marginally sensitive to prompt wording, particularly with stronger LLMs. More importantly, model behavior was driven by

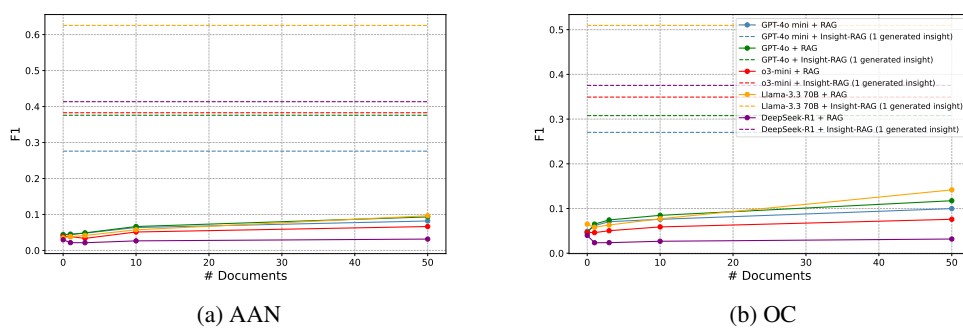

(a) AAN          (b) OC

Figure 7: The performance comparison of RAG versus Insight-RAG across the AAN and OC datasets based on F1 metric for deeply buried information. As demonstrated, Llama-3.3 performed the best, while DeepSeek-R1 performed the worst.

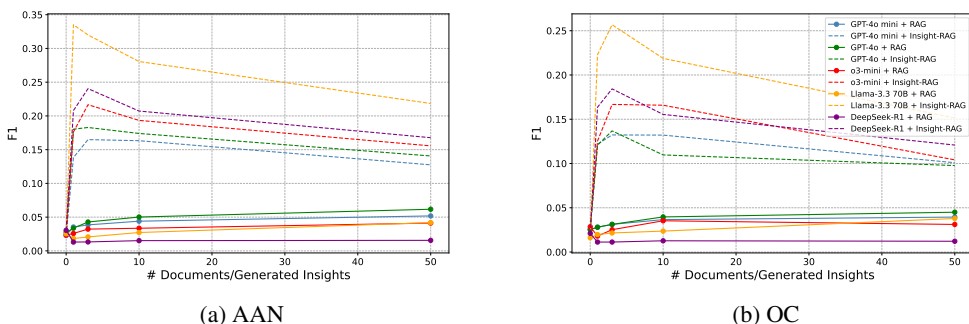

(a) AAN          (b) OC

Figure 8: The performance comparison of RAG versus Insight-RAG across the AAN and OC datasets based on averaged F1 metric for multi-document questions. As demonstrated, Llama-3.3 performed the best, while DeepSeek-R1 performed the worst.

the clarity and structure of the instructions rather than specific phrasings. When instructions were explicit and well-scoped, the models consistently demonstrated robustness to superficial prompt variations.

## C EXPERIMNETS

### C.1 F1 SCORE EVALUATION RESULTS

We report F1 and averaged F1 performance for all models for deeply buried and multi-document questions in Figure 7 and 8, respectively. Interestingly, despite DeepSeek's superior performance in Exact Match metrics, its F1 scores show a significant decline. Upon closer examination, we discovered this discrepancy stems primarily from DeepSeek's tendency to generate excessive content and occasional hallucinations, particularly when the correct document isn't retrieved. This poor F1 performances occur despite our removal of DeepSeek's "thinking" sections when calculating F1 scores. The other evaluated models demonstrate performance patterns similar to their Exact Match results, with Llama-3.3 70B consistently emerging as the top-performing model across both setting. Moreover, Table 5 presents the F1 scores for the paper matching task. While these results follow similar trends as the accuracy metric, the F1 scores reveal that both the positive and negative impacts of conventional RAG as well as the benefits of Insight-RAG, are even more amplified compared to accuracy.

| Model | AAN | | | OC | | |
|-------|-----|-----|-----|-----|-----|-----|
| | Vanilla | RAG (1 doc) | Insight-RAG | Vanilla | RAG (1 doc) | Insight-RAG |
| GPT-4o mini | 78.8 | 79.9 (+1.1) | 82.2 (+3.4) | 66.0 | 57.9 (-8.1) | 72.5 (+6.5) |
| GPT-4o | 82.4 | 77.6 (-4.8) | 82.8 (+0.4) | 61.2 | 66.3 (+5.1) | 65.6 (+4.4) |
| o3 mini | 85.0 | 85.1 (+0.1) | 85.4 (+0.4) | 70.4 | 65.4 (-5.0) | 78.9 (+8.5) |
| Llama 3.3 70B | 83.8 | 80.0 (-3.8) | 84.8 (+1.0) | 73.8 | 71.8 (-2.0) | 77.8 (+4.0) |
| DeepSeek-R1 | 59.3 | 66.7 (+7.4) | 68.6 (+9.3) | 50.4 | 60.6 (+10.2) | 62.2 (+11.8) |

Table 5: The F1 performance comparison of RAG versus Insight-RAG across the AAN and OC datasets in the paper matching task. As demonstrated, o3 mini performs the best while DeepSeek-R1 shows the lowest performance. Moreover, we observe that Insight-RAG consistently improves performance across all models, while RAG-based solutions show mixed impacts on model performance.

| Model | AAN | | | | OC | | | |
|-------|-----|-----|-----|-----|-----|-----|-----|-----|
| | 1 | 3 | 10 | 50 | 1 | 3 | 10 | 50 |
| GPT-4o (Deep) | 24.8 | 30.8 | 31.3 | 33.4 | 23.6 | 29.0 | 32.1 | 33.9 |
| DeepSeek-R1 (Deep) | 27.9 | 34.8 | 36.5 | 39.9 | 25.3 | 33.9 | 39.2 | 40.9 |
| GPT-4o (Multi) | 25.6 | 32.9 | 38.1 | 39.1 | 26.3 | 30.6 | 33.7 | 34.5 |
| DeepSeek-R1 (Multi) | 27.0 | 34.2 | 40.4 | 41.8 | 26.4 | 34.4 | 37.9 | 39.6 |

Table 6: Self-RAG exact match and averaged exact match performance on deeply buried and multi-document questions. We report results using the top 1, 3, 10, and 50 retrieved documents.

## C.2 SELF-RAG PERFORMANCE

To provide a stronger baseline, we evaluated Self-RAG on our benchmark. We adopt a training-free Self-RAG implementation that replaces learned reflection tokens with prompt-based LLM decisions for retrieval and verification, a strategy shown to be effective in previous works (Li et al., 2024; Chang et al., 2024). Since GPT-4o and DeepSeek-R1 achieved the strongest performance among proprietary and open LLMs in the standard RAG setting, we focus on these two models. The results for Self-RAG—reported in terms of exact match and averaged exact match for the deeply buried and multi-document tasks, respectively—are shown in Table 6. While Self-RAG substantially improves standard RAG, particularly when fewer documents are retrieved, it is still consistently and significantly outperformed by Insight-RAG. This highlights that even sophisticated frameworks like Self-RAG continue to suffer from the fundamental challenges we target, underscoring superiority of Insight-RAG in addressing these issues.

## C.3 COLBERT-BASED RAG PERFORMANCE

We report RAG-based exact match and averaged exact match performance using ColBERT as the retriever in Figures 9 and 10. The results closely mirror the performance trends observed with the GTE model, further reinforcing our core findings on the limitations of surface-level retrieval and the advantages of insight-driven retrieval.

## C.4 TRIPLE-BASED RAG PERFORMANCE

Focusing on DeepSeek-R1 due to its superior performance, we report its RAG-based results when, instead of retrieving documents, we retrieve triples from the set of all extracted triples for each dataset. Table 7 provides the exact match accuracy for the deeply buried information setting, along with the averaged exact match accuracy for the multi-document setting. We observe that while the model shows similar behavior to document-based RAG, using much less context—since a triple is much shorter than a document—it still falls significantly short compared to Insight-RAG performance. The overall gap between triple-based RAG and Insight-RAG underscores the shortcomings of conventional retrieval approaches and the complexity of resolving them.

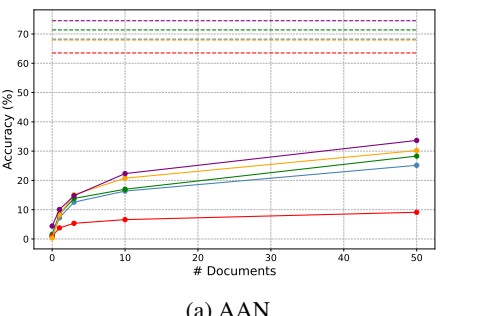
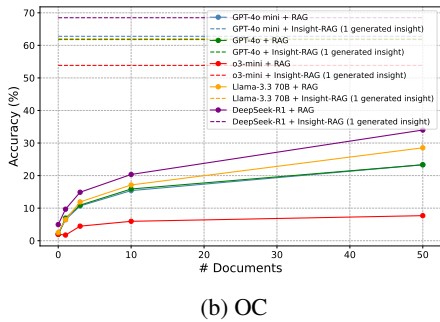

(a) AAN  (b) OC

Figure 9: We compare RAG and Insight-RAG on the AAN and OC datasets for questions based on deeply buried information using ColBERT as the retriever. DeepSeek-R1 performs best, followed by Llama-3.3 70B. Insight-RAG, even with a single generated insight, consistently outperforms RAG by a wide margin. Although retrieving more documents narrows the gap, Insight-RAG still maintains a clear advantage.

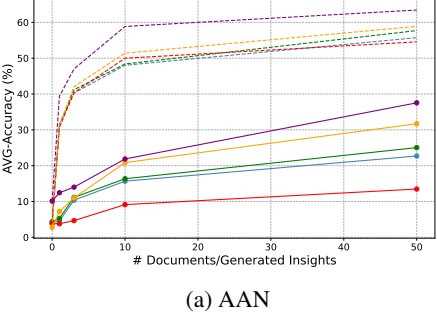
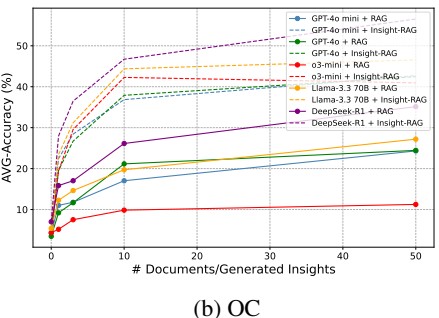

(a) AAN  (b) OC

Figure 10: We compare RAG and Insight-RAG on the AAN and OC datasets for multi-document questions using ColBERT as the retriever. DeepSeek-R1 performs best, followed by Llama-3.3 70B. Insight-RAG achieves much higher performance with just a few insights, with improvements slowing as more are added.

| Model | AAN | | | | OC | | | |
|---|---|---|---|---|---|---|---|---|
| | 1 | 3 | 10 | 50 | 1 | 3 | 10 | 50 |
| DeepSeek-R1 (Deep) | 13.8 | 18.9 | 25.8 | 35.2 | 20.1 | 27.0 | 33.0 | 42.2 |
| DeepSeek-R1 (Multi) | 12.1 | 14.0 | 14.7 | 25.2 | 10.6 | 13.9 | 17.9 | 22.7 |

Table 7: RAG-based exact match and averaged exact match accuracy of DeepSeek-R1 for deeply buried and multi-document questions. Instead of retrieving documents, we retrieve triples—using the set of extracted triples. We report results using the top 1, 3, 10, and 50 retrieved triples.

