# OpenReview forum: "Insight-RAG: Enhancing LLMs with Insight-Driven Augmentation"
_ICLR.cc/2026/Conference — Submitted to ICLR 2026_

### Official Review · Reviewer_pcc5 · 2025-10-31

**Soundness:** 2
**Presentation:** 1
**Contribution:** 2
**Rating:** 2
**Confidence:** 4

**Summary:**

The paper proposes a RAG framework (Insight-RAG) including an insight identifier, an insight retriever, and a response generator. The insight identifier involves a prompt for generating “sentence fragment that can be completed by the LLM trained on the dataset”. The insight retriever is essentially an LM fine-tuned in-domain to complete the sentence fragments. The question combined with the “insights” are inputs to the response generator, which produce the final answer. They also construct two new datasets by extracting triplets (subject, relation, object) from 5,000 papers, and converting the triplets into natural language questions. Results show that Insight-RAG is very effective in the proposed datasets.

**Strengths:**

1. The proposed method shows strong performances on the proposed dataset.

**Weaknesses:**

1. The experimental details are not clear. (1) It is not clear how the insight retriever is trained. How the retrieval is done is also unclear. From my understanding it’s just generating new tokens based on the insight, but then L372-373 mention recall @ 10. How do you sample 10 insights? (2) It is unclear how the train / dev / test split is constructed. Is the insight retriever trained on testing data? (3) How to extract triples from the papers is unclear. The prompt is not provided.
2. The datasets seem to be created for the method and are very synthetic. The results might not generalize to real-world datasets. Why not use existing RAG datasets? The fact that you fine-tuned on triplets, and the questions are constructed on the triplets is very concerning.
3. It is unclear where the performance gain comes from. It seems to me that most of the benefits come from fine-tuning the insight retriever on in-domain triplets (including those of the test data). The insight retriever is only memorizing the triplets and giving the correct objects during test time. If that is the case, I do not think this method is useful in practical scenarios. A quick way to find out if this is the case, you could fine-tuned an LM on the data you train the insight retriever on, and evaluate the performance of that LM with simple prompting. I think this should be a baseline to be included.
4. The presentation could be improved. (1) It is unclear from the main text what insights and questions look like. (I only understand when I look at the prompt in the Appendix.) It would be very helpful if you could include some examples in the main text. (2) Insight retriever is not actually a retriever. The naming should explain it clearly: it is simply a model fine-tuned in-domain. (3) Some results should not be in the appendix. For example, if you would like to refer to ColBERT and Self-RAG results, you should not put them in the appendix. (4) L176-L193 seem unnecessary.

**Questions:**

1. How is the insight retriever actually trained? What are the inputs and outputs? How do you train it on triplets? Is it trained on the triplets of the test data as well?
2. How do you construct train / development / test splits?
3. What is the prompt for extracting the triplets?
4. L214 you mentioned Recall @ k. How do you obtain top-K results?
5. L246-247 you mention the questions may be ambiguous or erroneous. Why even consider these questions? Why not filter them out?

---

### Official Review · Reviewer_fDW5 · 2025-11-01

**Soundness:** 2
**Presentation:** 2
**Contribution:** 2
**Rating:** 2
**Confidence:** 4

**Summary:**

The paper introduces Insight-RAG, a three-stage pipeline: (1) an LLM Insight Identifier converts a query/task into sentence-fragment insights; (2) an Insight Retriever (Llama-3.2-3B with LoRA continual pretraining on the target corpus and extracted triples) completes those fragments to surface content; (3) a final LLM generates the answer. The authors build bespoke benchmarks from AAN and OpenCorpus abstracts by extracting triples with GPT-4o-mini, translating them into questions, and evaluating deeply buried, multi-document, and a non-QA matching task. They report large gains over vanilla RAG and Self-RAG in some settings.

**Strengths:**

1. Clear articulation of perceived RAG failure modes (deeply buried facts, cross-doc synthesis, beyond-QA tasks).
2. Sensible modular decomposition and ablations on the two new components.
3. Some effort toward component analysis (identifier similarity scoring, retriever metrics) and token-budget sensitivity.

**Weaknesses:**

1. Benchmarks and labels are built from GPT-extracted triples taken from the same corpus used to continually pretrain the retriever, so training and evaluation are not cleanly separated.
2.Comparisons omit strong modern RAG setups (hybrid lexical+dense, reranking, trained Self-RAG/Iter-RetGen, HyDE, RePlug, etc.), making the reported large gains unconvincing.
3. The Insight Retriever appears to generate text rather than retrieve evidence/passages, making the comparison to passage-retrieval RAG apples-to-oranges and undermining attribution/grounding.
4. Synthetic, LLM-generated questions over abstracts only; brittle EM/F1 scoring without human adjudication or significance tests; constrained context budgets that handicap RAG.

**Questions:**

These are some suggestions-
1. Create document-level train/dev/test splits with strict exclusion of test docs (and their extracted triples) from CPT. Release the splits and scripts now.
2. Evaluate on established QA and multi-doc datasets (e.g., NQ-Open, HotpotQA, Multidoc2Dial, BioASQ/PubMedQA/LegalBench-RAG) and at least one real non-QA decision task with human labels.
3. Replace the Insight Identifier with query rewriting/keyword extraction; replace the Insight Retriever with (a) top-k passage retrieval + reranker; (b) CPT-only generator without the identifier; (c) non-parametric triple retrieval.

---

### Official Review · Reviewer_w3b3 · 2025-11-03

**Soundness:** 2
**Presentation:** 3
**Contribution:** 2
**Rating:** 4
**Confidence:** 3

**Summary:**

The authors propose Insight-RAG, introducing a dual intermediary insight extraction component in the RAG pipeline to address deeply buried information in knowledge bases.

**Strengths:**

- Comprehensive assessment of the methodology with multiple SOTA LLMs
- Authors provided a computational complexity analysis with cost implications and prompt sensitivity

**Weaknesses:**

- Lack of methodology motivation and benchmark choice explanations with respect to the paper objective (retrieve deeply buried information) - which otherwise remains overall vague.
- The authors claim that the contribution is represented by the coupling between the identifier and the retriever, yet these appear to be separately “prompted/trained” components
- Triples not defined, examples would be welcome
- (nit) Figure 6 does not contribute much in itself - Z scores are not very readable, could be moved in text / table

**Questions:**

- Are there other relevant RAG variants worth including in the comparison?
- How does the proposal compares to a fine tuned reranker? With respect to ColBERT, Figure 9 and 10 could benefit from direct comparison with the authors methodology (main text)
- What is the rationale of CPT for an insight retriever? Why not other (SFT, RL) approaches?

**Details Of Ethics Concerns:**

/

---

### Official Review · Reviewer_fcRV · 2025-11-03

**Soundness:** 2
**Presentation:** 3
**Contribution:** 2
**Rating:** 4
**Confidence:** 3

**Summary:**

Traditional retrieval mechanisms often fail to capture nuanced insights required for complex tasks and may neglect relevant insights dispersed across multiple documents. To deal with this problem, the authors propose a new method, Insight-RAG, that firstly identifies necessary insights to solve a task and then feeds the identified insights to a large language model (LLM). Experimental results on the newly created benchmark based on conventional two scientific paper datasets show that Insight-RAG can outperform strong baselines in accuracy with much less contextual information.

**Strengths:**

- The proposed method, Insight-RAG, contains components prepared specifically for capturing nuanced insights and handling multiple documents, addressing the fundamental issues in the conventional RAG.
- The authors created a new benchmark based on two conventional scientific paper datasets.
- The experiments use various large language models (LLMs).
- The experimental results show the large gains by using Insight-RAG.
- The analysis shows the contribution of each component.

**Weaknesses:**

- Even though Insight Retriever is fine-tuned by LoRA, the RAG baselines use retrievers without fine-tuning. This is an unfair comparison.
- The datasets used in the experiment are restricted to the scientific paper domain.
- The increased runtime caused by Insight Retriever is not reported.

**Questions:**

Does Insight Retriever work without fine-tuning?

---

### Meta-Review · Area_Chair_AB13 · 2026-01-07

**Summary:**

This paper proposes Insight-RAG, a 3-stage pipeline: (1) an LLM Insight Identifier turns the query into “insight fragments” (information requirements), (2) an Insight Retriever (actually an in-domain LM, CPT/LoRA on the corpus + extracted triples) completes/mines these fragments to produce “insights”, and (3) a final LLM answers using query + mined insights. They introduce new synthetic benchmarks built from two scientific-paper datasets, targeting: buried facts within a doc, cross-doc synthesis, and a non-QA matching-style task. They report very large gains (up to ~60 points) vs “traditional RAG”.

**Reviewer Concerns:**

Addressed: limited; some cost/complexity analysis exists (w3b3), some component ablations.

Outstanding: (1) clean doc-level splits + strict exclusion of test docs/triples from retriever training; (2) clear description of insight retriever training + how top-k/recall@k is computed; (3) stronger and fairer baselines (fine-tuned retriever/reranker, hybrid retrieval, HyDE/iterative retrieval, etc.); (4) validation on established RAG datasets and/or human evaluation; (5) runtime/cost reporting for the full pipeline.

**Reviewer Scores:**

No evidence here that rebuttal occurred / fixed issues. I’d assume:
- fDW5 stays 2 unless leakage + baselines are fixed.
- pcc5 stays 2 unless training/splits/top-k clarified and leakage ruled out.
- fcRV might move to 5–6 if they show (a) no fine-tuning advantage over tuned baselines, (b) clean splits, (c) runtime accounted.
- w3b3 might move to 5 with better motivation, clearer definitions/examples, and comparisons to rerankers/ColBERT-style setups.

---

### Decision · Program_Chairs · 2026-01-26

Reject